# Multivariate Linear Regression Models to Predict Monomer Poisoning Effect in Ethylene/Polar Monomer Copolymerization Catalyzed by Late Transition Metals

**Wei Zhao [1], Zhihao Liu [1], Yanan Zhao [1,*], Yi Luo [1,2,*] and Shengbao He [2]**

[1]  State Key Laboratory of Fine Chemicals, School of Chemical Engineering, Dalian University of Technology, Dalian 116024, China; weizhao3494@163.com (W.Z.); zhihao-liu@outlook.com (Z.L.)
[2]  PetroChina Petrochemical Research Institute, Beijing 102206, China; hsb@petrochina.com.cn
[*]  Correspondence: yananzhao@dlut.edu.cn (Y.Z.); luoyi@dlut.edu.cn (Y.L.)

**Abstract:** This study combined density functional theory (DFT) calculations and multivariate linear regression (MLR) to analyze the monomer poisoning effect in ethylene/polar monomer copolymerization catalyzed by the Brookhart-type catalysts. The calculation results showed that the poisoning effect of polar monomers with relatively electron-deficient functional groups is weaker, such as ethers, and halogens. On the contrary, polar monomers with electron-rich functional groups (carbonyl, carboxyl, and acyl groups) exert a stronger poisoning effect. In addition, three descriptors that significantly affect the poisoning effect have been proposed on the basis of the multiple linear regression model, viz., the chemical shift of the vinyl carbon atom and heteroatom of polar monomer as well as the metal-X distance in the σ-coordination structure. It is expected that these models could guide the development of efficient catalytic copolymerization system in this field.

**Keywords:** density functional theory; multivariate linear regression; poisoning effect of polar monomers; Brookhart-type catalysts

## 1. Introduction

Compared with the polyolefins, the incorporation of polar monomers into nonfunctionalized polyolefin backbones can significantly improve various properties of polymers, such as flexibility, adhesion, protective properties, surface properties, solvent resistance, which leads to expanding the range of applications [1–6]. It is well known that metal-catalyzed coordination-insertion copolymerization of olefinic hydrocarbons with polar monomers is the most convenient and economical synthetic strategy. Generally, the early-transition-metal complexes with high oxygen affinity are easily poisoned by polar functional groups [7]. It is, therefore, necessary to use late-transition-metal complexes (Ni or Pd) with low oxygen affinity to catalyze the coordination copolymerization of polar monomers.

In this context, various late-transition-metal catalysts have been developed (Figure 1a) [8,9]. In the mid-1990s, a groundbreaking work was achieved by using cationic Ni/Pd catalysts based on α-diimine ligands by Brookhart (**II** in Figure 1a,) [10]. Since then, a series of complexes [11–14] have been developed on the basis of the Brookhart-type catalysts, which are only suitable for a small part of simple monomers such as acrylates [15], vinyl ketones [10], and silyl vinyl ethers [16–19]. In 2002, Drent-type catalysts were reported by Drent and co-workers (**III** in Figure 1a,) [20]. In addition, they expanded the scope of substrate for copolymerization, such as vinyl fluoride [21], vinyl ethers [22], and some important methylene-spaced polar monomers (with a spacer between the polar group and the double bond) [23–28]. However, there are still some obvious disadvantages in these catalytic systems: low copolymerization activity, low insertion rate and low molecular weight [29], etc. Among them, the low copolymerization activity is a common challenge

for these catalysts. It is noteworthy that the main reason for the low activity of copolymerization is the occurrence of the poisoning effect. The tolerance of the same catalyst to different functional groups is obviously different, but there is a lack of systematic study on this difference. Therefore, systematically exploring the poisoning effect of polar monomers is helpful for improving the copolymerization activity. Herein, multivariate linear regression analyses on the poisoning effect of different polar monomers have been conducted based on DFT calculations at the molecular level, taking the conventional Brookhart-type catalysts **II** as a model. Various polar monomers [30–32] and the copolymerization mechanism are shown in Figure 1 and Figure S1. When $\Delta\Delta E(\pi\text{-}\sigma) < 0$, the $\pi$-complex (double bond coordination) is more stable than the heteroatom coordination complex ($\sigma$-complex), and vice versa. As shown in Figure 1c, if the $\sigma$-complex (**B2**) is more stable than the vinyl-coordination $\pi$-complex (**B3**), the catalyst could be poisoned and inactive. Through multiple linear regression analysis, it has been demonstrated that the chemical shifts of a vinyl carbon atom ($^{mon}NMR_C{}^{\beta}$) and the coordinating heteroatom ($^{mon}NMR_X$) of polar monomers, as well as the metal-heteroatom distance ($^{B2}bond_{Pd/Ni\text{-}X}$), are the key factors governing the poisoning effect. The current prediction models are expected to be useful for the prediction of the poisoning effect of other polar monomers in the polymerization catalyzed by Brookhart-type catalysts.

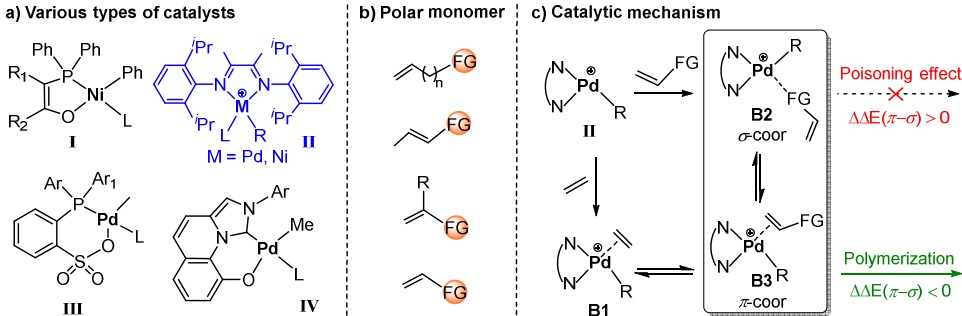

**Figure 1.** (**a**) Four conventional catalyst structures [8–10,20]. (**b**) Several different polar monomer structures. (**c**) The copolymerization mechanism of ethylene/polar monomer meditated by Brookhart-type catalysts **II**.

## 2. Computational Details

All DFT calculations were performed with Gaussian 16 program [33]. The D3 [34] dispersion-corrected density functional method B3LYP functional [35–37], together with the 6-311G(d) basis set for nonmetal atoms (C, H, O, N, P, S, F, Si and Cl) and the LANL2DZ [38–40] basis set, as well as the associated poseupotential for metal atoms (Pd and Ni) was used for geometry optimizations. All optimizations were carried out in the gas-phase. The noncovalent interaction (NCI) analysis [41] by Multiwfn [42] and VMD [43] softwares was carried out for important structures. The optimized tridimensional geometrical structures were represented by CYLView [44]. Color map and multiple linear regression analysis were performed with Matlab program. Taking diimide palladium as an example, single-point calculations were further performed at the higher level by using the density functional method M06 [45]; 6-311 + G (d, p) was used for the nonmetal atoms; the basis set LANL2DZ [38–40], as well as the associated pseudopotential, were applied for the Pd atom. In these single-point calculations, the solvation effect of toluene ($\varepsilon = 2.37$) was considered through the CPCM model [46,47]. It can be seen from comparison of results by solvated and gas-phase that the effect of solvation exerted a minor effect on the trend of poisoning effects (see Figure S1).

## 3. Results and Discussion

The poisoning effect of several different types of polar monomers (see Figure S2) were explored by DFT calculation and multiple linear regression analysis. Generally, the energy difference $\Delta\Delta E(\pi\text{-}\sigma)$ (Figure 1c) between double bond coordination and heteroatom

coordination is directly related to the poisoning effect. Firstly, the coordination process of a series of polar monomers catalyzed by complex $II_{Pd}$ and $II_{Ni}$ were calculated. Through theoretical calculation, it was found that the energy difference of some polar monomers was too high ($\Delta\Delta E(\pi\text{-}\sigma) > 0$), such as acrylonitrile, methylene spacer vinyl and internal olefin monomers; thus, there are very obvious poisoning effects. On the contrary, the poisoning effect of ether olefins and halogenated olefins is relatively weaker theoretically ($\Delta\Delta E(\pi\text{-}\sigma) < 0$, Figure 2). However, although the poisoning effect of halogenated olefins is weak (Figure 2a), the $\beta$-halide or or $\beta$-OR elimination reaction can easily occur in the polymerization, resulting in low polymerization activity [48,49]. For polar monomers 42, 56, 57, this is a kind of styrenic monomers, which are easy to coordinate metal to produce stable and inactive $\eta^3$-complexes [50]. These results are consistent with the experimental phenomena [51]. Moreover, the number of polar monomers that cannot poison Ni complexes (Figure 2a) is significantly less than that of Pd complexes (Figure 2a,b).

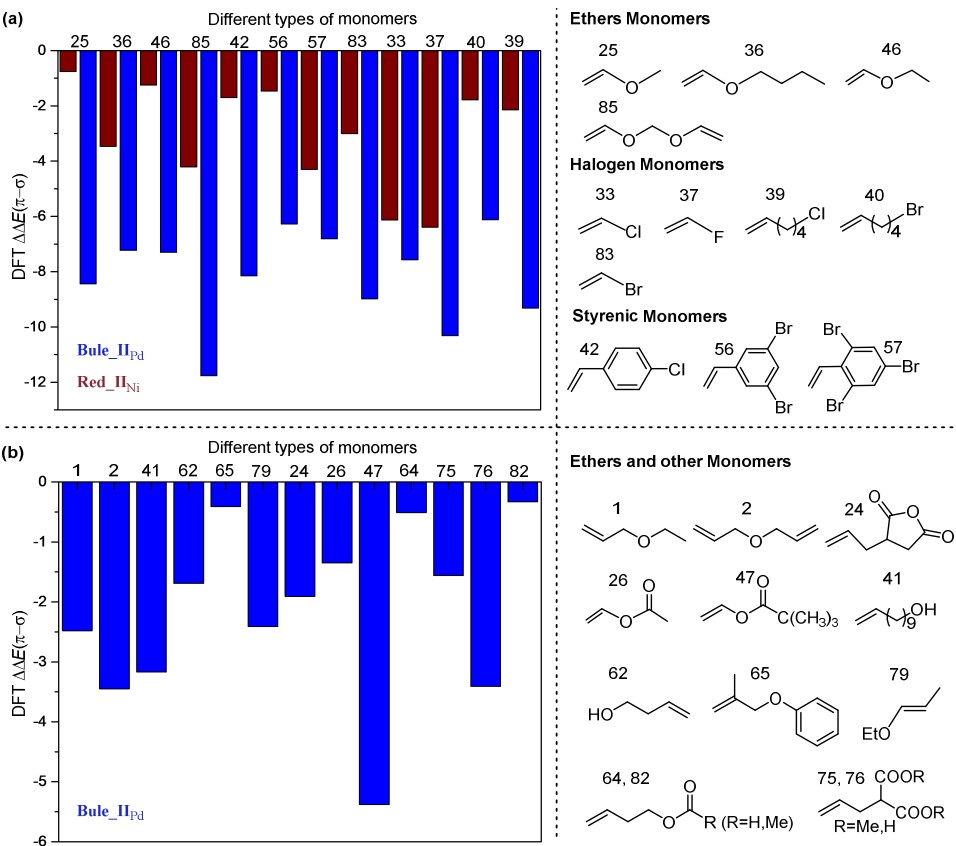

**Figure 2.** (**a**) The polar monomers with weak poisoning effect catalyzed by $II_{Ni}$ and $II_{Pd}$ complexes ($\Delta\Delta E(\pi\text{-}\sigma) < 0$). (**b**) Other monomers with weak poisoning effect catalyzed by $II_{Pd}$ complex.

Before the MLR analysis, it is necessary to calculate the electronic and stereoscopic descriptors of different polar monomers by DFT calculation. At present, as hundreds of molecular descriptors are available in the literature [42–54], we selected a number of them based on our previous experience on olefin polymerization [55–58] and the literature about MLR analysis of transition-metal-based reactivity. For the predictive effect and the convenience of the MLR model, only one of heteroatom coordination and double bond coordination should be selected for descriptor calculation. Firstly, the descriptors of heteroatom coordination structure **B2** and polar monomers are calculated for multivariate linear regression. A total of 21 descriptors were calculated, including Sterimol values [59] ($^{B2}B1_{Ni/Pd\text{-}X}$, $^{B2}B5_{Ni/Pd\text{-}X}$ and $^{B2}L_{Ni/Pd\text{-}X}$), steric hindrance of metal center ($^{B2}Steric_{Ni/Pd}$), bond length ($^{B2}bond_{Ni/Pd\text{-}X}$), LUMO ($^{B2}LUMO$), dihedral angle ($^{B2}\angle XNi/PdN_1C_1$ and $^{B2}\angle C_3Ni/PdN_2C_2$), Infrared freq ($^{mon}IR_{C=C}$) and Freq Intensities ($^{mon}v_{C=C}$), NMR ($^{mon}NMR_C{}^{\alpha}$, $^{mon}NMR_C{}^{\beta}$ and $^{mon}NMR_X$),

NBO ($^{B2}NBO_{Ni/Pd}$, $^{mon}NBO_X$, $^{mon}NBO_C^\alpha$ and $^{mon}NBO_C^\beta$), Polarizability ($^{mon}\alpha$), HOMO ($^{mon}HOMO$), volume ($^{mon}V$) and Dipole Moment ($^{mon}\mu$). Having both computed $\Delta\Delta E(\pi\text{-}\sigma)$ and descriptors in hand, we performed univariate correlation analysis for the whole data set (87 reactions) to investigate the relationship between descriptors and $\Delta\Delta E(\pi\text{-}\sigma)$, and to see the variation trends in poisoning effect (by complex $\mathbf{II_{Ni}}$). For this purpose, a correlation matrix for the selected parameters was generated, as represented by a color map [60] (Figure 3). As shown in Figure 3, relatively strong correlations were found between $\Delta\Delta E(\pi\text{-}\sigma)$ and some electronic parameters involving $\mathbf{D_2}$ ($^{mon}NMR_C^\beta$, $|R| = 0.76$), $\mathbf{D_3}$ ($^{mon}NMR_X$, $|R| = 0.59$), $\mathbf{D_5}$ ($^{mon}NBO_X$, $|R| = 0.60$) and $\mathbf{D_{15}}$ ($^{B2}bond_{Ni\text{-}X}$, $|R| = 0.70$). These descriptors with high correlation coefficient $|R|$ can presumably exert a major influence on $\Delta\Delta E(\pi\text{-}\sigma)$. Conversely, steric hindrance descriptors exert little effect on $\Delta\Delta E(\pi\text{-}\sigma)$, such as $\mathbf{D_{17}}$–$\mathbf{D_{21}}$.

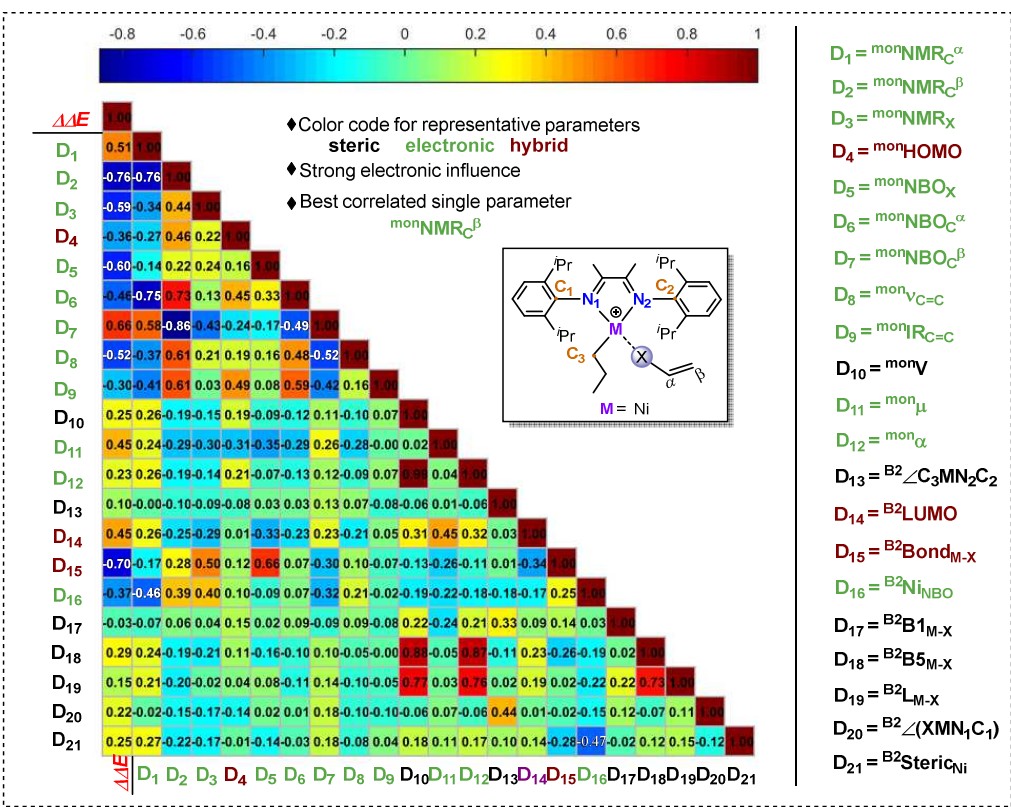

**Figure 3.** Correlation color map. The first column corresponds to the single-parameter correlations of the $\Delta\Delta E(\pi\text{-}\sigma)$, while the others represent the interparameter correlations [60].

According to the color map in Figure 3, we selected four single parameters $\mathbf{D_2}$ ($^{mon}NMR_C^\beta$, $|R| = 0.76$), $\mathbf{D_3}$ ($^{mon}NMR_X$, $|R| = 0.59$), $\mathbf{D_5}$ ($^{mon}NBO_X$, $|R| = 0.60$) and $\mathbf{D_{15}}$ ($^{B2}bond_{Ni\text{-}X}$, $|R| = 0.70$) with relatively higher absolute values of R to performed univariate correlation analysis to investigate the relationship between descriptors and $\Delta\Delta E(\pi\text{-}\sigma)$ and to see the variation trends of poisoning effect. However, when we evaluated the full set of data (87 reactions) in Figure S1, the single parameters cannot describe well the trend of the $\Delta\Delta E(\pi\text{-}\sigma)$ value ($R^2 < 0.60$, Figure 4), suggesting that multivariate linear regression analysis was required to describe the combined multivariate influences of polar monomers on poisoning effect.

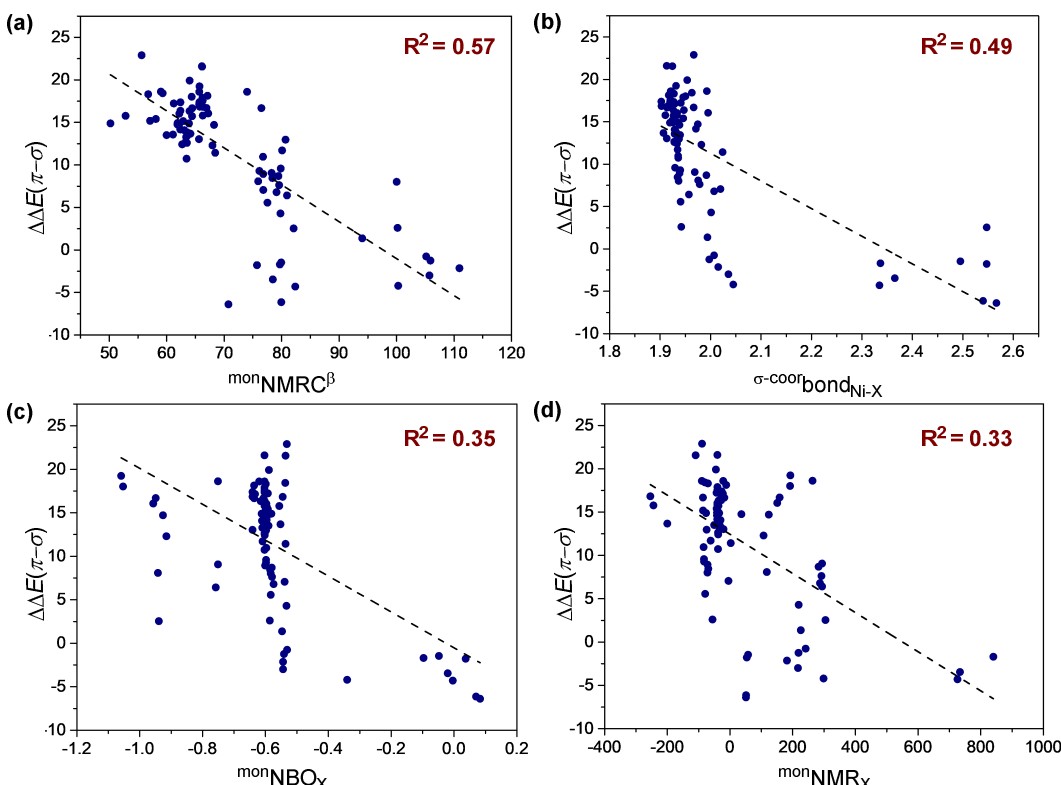

**Figure 4.** Representative univariate trends. Different polar monomers catalyzed by $II_{Ni}$ complex.

Next, Linear regression modeling was applied to correlate the poisoning effect (expressed as $\Delta\Delta E(\pi\text{-}\sigma)$) to the above calculated descriptors. A total of 87 reactions were used in this paper, of which 60 reactions were randomly selected as the training set and 27 reactions as the test set. Considering that the accuracy of the simplified regression model can be expressed more clearly via a graphical manner, a plot of the calculated $\Delta\Delta E(\pi\text{-}\sigma)$ values and that predicted by this model is depicted in Figure 5. When complex $II_{Ni}$ was used, the results demonstrated a high correlation ($R^2 = 0.85$, $Q^2 = 0.83$; $Q^2$ represents the value of leave-one-out cross validation and showed that there was no overfitting) between predicted $\Delta\Delta E(\pi\text{-}\sigma)$ values and calculated ones (Figure 5a). Moreover, when complex $II_{Pd}$ is used, the result shows a relatively lower correlation between predicted $\Delta\Delta E(\pi\text{-}\sigma)$ values and calculated ones ($R^2 = 0.79$, $Q^2 = 0.78$, Figure 5b). Notably, as the models were acquired from normalized descriptors, the resulting coefficients can indicate the significance of the represented descriptor. Therefore, it can be seen from the model that, for complex $II_{Ni}$ and $II_{Pd}$ systems, the descriptors $^{mon}NMR_C{}^\beta$ and $^{B2}Bond_{Ni/Pd\text{-}X}$ exert a significant impact on the poisoning effect. Nevertheless, $^{mono}NMR_X$ holds a relatively weaker effect on the poisoning effect of $II_{Ni}$ systems and almost no effect on $II_{Pd}$ systems. Besides, the descriptors of $\pi$-coordination structure **B3** (in complex $II_{Ni}$ systems) and polar monomers are also calculated (Figure S3), but the prediction results ($R^2 = 0.82$) based on these descriptors are not as satisfactory as the above results ($R^2 = 0.85$, Figure 5a).

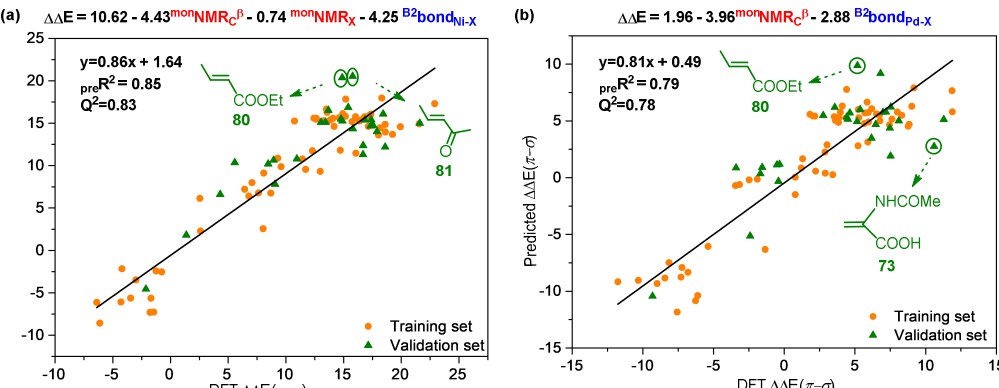

**Figure 5.** Plot of computed vs. predicted $\Delta\Delta E$(π-σ) (kcal/mol) for complexes **II$_{Ni}$** (**a**) and **II$_{Pd}$** (**b**) using the multivariate linear regression models.

It is worth noting that there were several deviation points in both complexes **II$_{Ni}$** (a) and **II$_{Pd}$** (b) systems (points in green circle, Figure 5a,b), and these deviation points may reduce the prediction accuracy of the MLR model; thus, it is necessary to analyze the origins of these deviation points. Through structural analysis, it was found that these deviation points share something in common, that is, there are obvious non-covalent weak interactions in the σ-coordination structure (Figure 6), including hydrogen bond, H···π interactions and heteroatom···π interactions. In order to confirm the effect of non-covalent weak interaction on energy of σ-coordination structure, the aromatic ring on the ligand was substituted with methyl, and the $\Delta\Delta E$(π-σ) was calculated (Figure S4). By comparing the calculation results, it can be seen that the non-covalent weak interaction between polar monomers and aromatic ring on ligands exerted an obvious effect on the energy of $\Delta\Delta E$(π-σ), but the non-covalent weak interaction is not considered in above descriptors. Therefore, there is a certain deviation in the prediction results of the system with obvious non-covalent weak interaction. Furthermore, external validation of the MLR model for the complex **II$_{Ni}$** (a) and **II$_{Pd}$** were performed separately; results are shown in Figure 7. These results indicate that these MLR models have certain extrapolation and prediction ability. It is noted that some monomers containing an active hydrogen or a strong electrophilic group could undergo side reactions in the polymerization system rather than polymerize. However, these monomers were considered for expending the scope of data set and were helpful for constructing the prediction models [61–63]. It is also noteworthy that there may be secondary interactions between polar monomer molecules in the polymerization system [4,64]. Our work mainly focuses on the influence of the electronic or stereoscopic effect of the monomer itself on the poisoning effect of the catalyst. The actual experimental reaction system was very complex; there will be a variety of interactions. In order to explore the main reasons affecting the poisoning effect, this work simplifies the experimental conditions and provides a valuable reference for experimenters to screen suitable polymerized monomers.

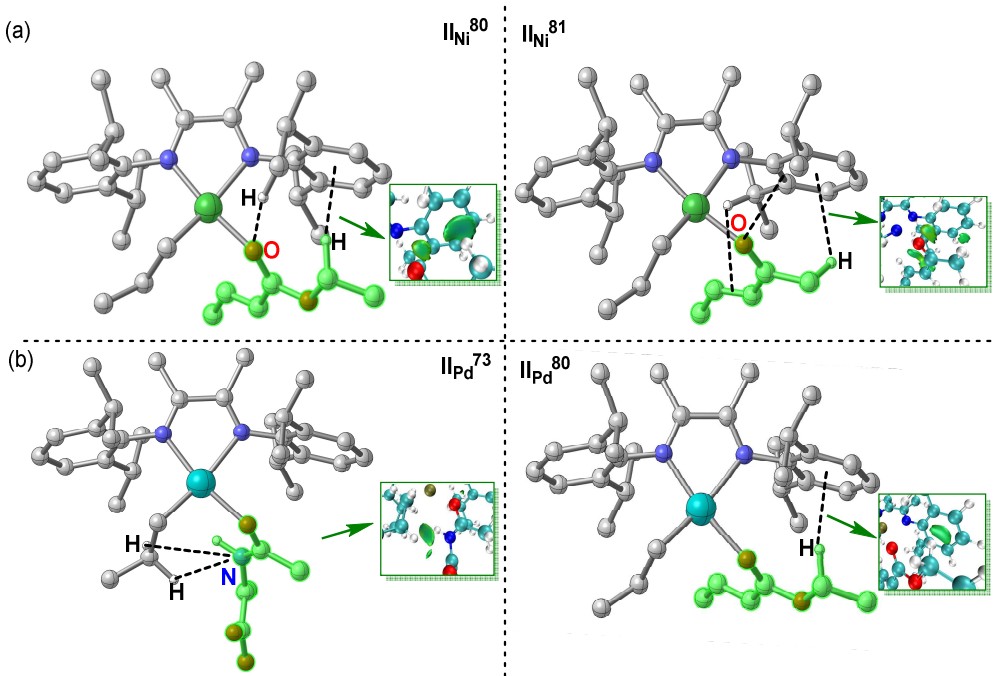

**Figure 6.** Structure and non-covalent interaction of deviation points in complexes **II$_{Ni}$** (**a**) and **II$_{Pd}$** (**b**) systems.

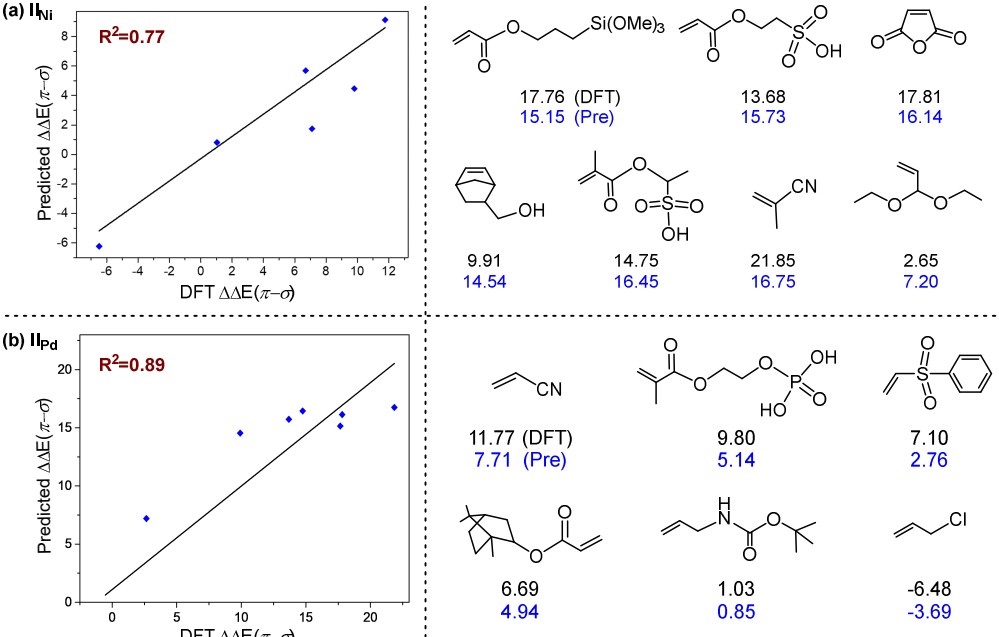

**Figure 7.** External verification of the multivariate linear regression models for complexes **II$_{Ni}$** (**a**) and **II$_{Pd}$** (**b**). Blue represents predicted $\Delta\Delta E(\pi\text{-}\sigma)$ (kcal/mol), black represents computed $\Delta\Delta E(\pi\text{-}\sigma)$ (kcal/mol).

## 4. Conclusions

In summary, the poisoning effect of different polar monomers by Brookhart-type catalyst were explored through the combination of DFT calculations and multiple linear regression analyses. The results show that the combination of the structure of heteroatom-coordination complex and the descriptors of polar monomer can establish a relationship between the structure and the poisoning effect represented by $\Delta\Delta E(\pi\text{-}\sigma)$. It is found that the

descriptors of $^{mon}NMR_C{}^{\beta}$ and $^{B2}Bond_{Ni/Pd-X}$ are the key factors affecting the poisoning effect. In addition, in the case of the Ni system, the $^{mon}NMR_X$ descriptor also plays a certain role in the poisoning effect. Besides, by analyzing the σ-coordination structure of the deviation point, it was found that the non-covalent interaction between polar monomers and the catalyst may be the main reason for the deviation of the predicted value by the multiple linear regression model. Moreover, the result of external verification shows that such prediction models possess a certain ability to predict and extrapolate the poisoning effect of other polar monomers. Such a combination of DFT-derived energy difference and multidimensional quantitative analysis is expected to be effective in assessing the other polymerization performance and can provide new and efficient monomer screening strategies for experimentalists.

**Supplementary Materials:** The following supporting information can be downloaded at: https://www.mdpi.com/article/10.3390/inorganics10020026/s1, Figure S1: Gas-phase vs. solvation effect $\Delta\Delta E(\pi\text{-}\sigma)$ (kcal/mol) for complex IIPd; Figure S2: structures of polar monomers; Figure S3: plot of computed vs. predicted $\Delta\Delta E(\pi\text{-}\sigma)$ (kcal/mol) for complex IINi (base on B3 structure) using the multivariate linear regression models; Figure S4: changes of structure and $\Delta\Delta E(\pi\text{-}\sigma)$ after substitution of aromatic ring on catalyst (complex IINi) with methyl. Electronic energy data of polar monomers coordination, polar monomers and other structures; and electronic and stereoscopic descriptors of polar monomers, complexes B2 and B3 (Excel) (XLSX). Optimized stationary points (ZIP).

**Author Contributions:** Conceptualization, Y.L., Y.Z. and S.H.; methodology, W.Z., Y.Z. and Z.L.; investigation, W.Z. and Y.Z.; data curation, W.Z., Z.L. and Y.Z.; writing—original draft preparation, W.Z. and Y.Z.; writing—review and editing, Y.L., Y.Z. and W.Z.; funding acquisition, Y.L., Y.Z. All authors have read and agreed to the published version of the manuscript.

**Funding:** This research was funded by the NSFC (Nos. 22071015, U1862115, and 22101041. China Postdoctoral Science Foundation (2021M700664)).

**Institutional Review Board Statement:** Not applicable.

**Informed Consent Statement:** Not applicable.

**Data Availability Statement:** This research did not report any data.

**Acknowledgments:** This work was supported by the NSFC (Nos. 22071015, U1862115, and 22101041, China Postdoctoral Science Foundation (2021M700664)). The authors also thank the Network and Information Center of Dalian University of Technology for part of computational resources.

**Conflicts of Interest:** The authors declare no conflict of interest. The funders had absolutely no role in the design of the study; in the collection, analyses, or interpretation of data; in the writing of the manuscript, or in the decision to publish the results.

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
