# Peer review of "Multivariate Linear Regression Models to Predict Monomer Poisoning Effect in Ethylene/Polar Monomer Copolymerization Catalyzed by Late Transition Metals"

_inorganics, doi:10.3390/inorganics10020026_

Round 1

Reviewer 1 Report

In this manuscript Luo and coworkers report on combining the DFT calculations and multivariate linear regression to understand the monomer poisoning effect on ethylene/polar monomer copolymerization using Brookhart type catalysts. The manuscript is well written and results are well explored by choosing different monomers. The manuscript can be accepted for publication in Inorganics  after addressing the effect of solvation  on the obtained results.

Reviewer 2 Report

This paper describes multivariate linear regression models to predict monomer poisoning effect in ethylene/polar monomer copolymerization catalyzed by alpha-diimine Ni/Pd catalysts. The calculation results are interesting, and can guide copolymerization of ethylene and what kind of polar monomer. I recommend publication of this manuscript after minor revisions. (1) For copolymerizations of ethylene and polar monomer using alpha-diimine Ni/Pd catalysts, some literatures should be cited. For example: Macromolecules 2017, 50, 2675–2682 (2) In Figure 1, structure of [P,O] catalysts (I) may be incorrect, please check it. (3) In page 2, sigma or pi are missed in the text. (4) For polar monomers 42, 56, 57, this is a kind of styrenic monomers, which are easy to coordinate metal to produce stable and inactive eta3 complexes (see Macromolecules 2020, 53, 256–266). I suggest that they should be listed as a kind of styrenic monomers.

Reviewer 3 Report

In this manuscript, the authors explored the toxic effects of different polar monomers on Brookhart-type catalysts by combining density functional theory calculations with multivariate linear regression analysis. A relationship between the structure and the poisoning effect represented by ΔΔE(π-σ) was proposed based on the combination of the structure of heteroatom-coordination complex and the descriptors of polar monomer. Moreover, the authors believe that the prediction model has a certain ability to predict and extrapolate the poisoning effects of other polar monomers. The authors' study is detailed and the point is clear, so it's worth publishing on Inorganics. As an experimental researcher in the field of olefin polymerization, my major concern with this manuscript comes from the gap between theory and practice. First, many of the polar monomers considered in the manuscript do not actually exist in the polymerization system in the form of the structural formula drawn. For example, a monomer containing an active hydrogen or a strong electrophilic group can react with an alkylaluminum-based cocatalyst used in the α-diimine Ni catalysis or the catalyst itself (J. Am. Chem. Soc. 2002, 124, 1176-1177) (Nat. Commun. 2021, 12, 6283) ( J. Polym. Sci., Polym. Chem. 1999, 37, 2457−2469). Second, there may be secondary interactions between polar monomer molecules (Macromolecules 2018, 51, 6818–6824) (J. Am. Chem. Soc. 2022, doi: 10.1021/jacs.1c11817). Finally, the solvent effect may have a significant effect on the properties of olefin polymerization catalysts (J. Am. Chem. Soc. 2018, 140, 6685−6689) (Nat. Catal. 2019, 2, 236–242), which is not considered in the manuscript.

Author Response

The file of reviewer 2 was not uploaded due to a mistake. You can see the replies to reviewers 2 and 3 here. Please see the attachment!
